# DEEP DATA FLOW ANALYSIS

## ABSTRACT

Compiler architects increasingly look to machine learning when building heuristics for compiler optimization. The promise of automatic heuristic design, freeing the compiler engineer from the complex interactions of program, architecture, and other optimizations, is alluring. However, most machine learning methods cannot replicate even the simplest of the abstract interpretations of data flow analysis that are critical to making good optimization decisions. This must change for machine learning to become the dominant technology in compiler heuristics.

To this end, we propose PROGRAML – *Program Graphs for Machine Learning* – a language-independent, portable representation of whole-program semantics for deep learning. To benchmark current and future learning techniques for compiler analyses we introduce an open dataset of 461k Intermediate Representation (IR) files for LLVM, covering five source programming languages, and 15.4M corresponding data flow results. We formulate data flow analysis as an MPNN and show that, using PROGRAML, standard analyses can be learned, yielding improved performance on downstream compiler optimization tasks.

## 1 INTRODUCTION

Compiler implementation is a complex and expensive activity (Cooper & Torczon, 2012). For this reason, there has been significant interest in using machine learning to automate various compiler tasks (Allamanis et al., 2018). Most works have restricted their attention to selecting compiler heuristics or making optimization decisions (Ashouri et al., 2018; Wang & O'Boyle, 2018). Whether learned or engineered by human experts, these decisions naturally require reasoning about the program and its behavior. Human experts most often rely upon *data flow* analyses (Kildall, 1973; Kam & Ullman, 1976). These are algorithms on abstract interpretations of the program, propagating information of interest through the program's control-flow graph until a fixed point is reached (Kam & Ullman, 1977). Two examples out of many data flow analyses are: *liveness* – determining when resources become dead (unused) and may be reclaimed; and *available expressions* – discovering which expressions have been computed on all paths to points in the program. Prior machine learning works, on the other hand, have typically represented the entirety of the program's behavior as a fixed-length, statically computed feature vector (Ashouri et al., 2018). Typical feature values might be the number of instructions in a loop or the dependency depth. The weakness of these techniques is shown by the fact that they are trivially confused by the addition of dead code, which changes their feature vectors without changing the program's behavior or its response to optimizations. Such learning algorithms are unable to learn their own abstract interpretations of the program and so cannot avoid these pitfalls or more subtle versions thereof (Barchi et al., 2019).

Recently, there have been attempts to develop representations that allow finer-grain program reasoning. Many, however, are limited both by how inputs are represented as well as how inputs are processed. Representations based on source code and its direct artifacts (e.g., AST) (Alon et al., 2018a; Yin et al., 2018; Haj-Ali et al., 2020) put unnecessary emphasis on naming and stylistic choices that may not correlate with the functionality of the code (e.g., Fig. 2a). Approaches based on intermediate representations (IR) (Ben-Nun et al., 2018; Mirhoseini et al., 2017; Brauckmann et al., 2020) remove such noise but fail to capture information about the program that is important for analysis (e.g., Fig. 2b variables, Fig. 2c commutativity). In both cases, models are expected to reason about the flow of information in programs using representations that do not directly encode this information. Clearly, a program representation is needed that enables machine learning algorithms to reason about the execution of a program by developing its own data flow analyses.

Figure 1: Our proposed approach for compiler analyses driven by graph-based deep learning.

Since current approaches are ill-suited to program-wide data flow analysis, we propose overcoming their limitations by making the program's control, data, and call dependencies a central part of the program's representation *and* a primary consideration when processing it. We achieve this by seeing the program as a graph in which individual statements are connected to other statements through relational dependencies. Each statement in the program is understood only in the context of the statements interacting with it. Through relational reasoning (Battaglia et al., 2018), a latent representation of each statement is learned that is a function of not just the statement itself, but also of the (latent) representations of its graph neighborhood. Notably, this formulation has a striking similarity to the IRs used by compilers, and the iterative propagation of information resembles the *transfer functions* and *meet operators* in traditional data flow analyses (Kildall, 1973).

Recently proposed techniques for learning over graphs have shown promise in a number of domains (Schlichtkrull et al., 2018; Ziwei et al., 2020). With a suitable representation and graph-based model, we extend these approaches to the domain of compiler analysis, enabling downstream tasks built on top of such graph models to natively incorporate reasoning about data flow into their decision making. This improves performance on downstream tasks without requiring additional features.

We make the following contributions:

- We propose a portable, language-independent graph representation of programs derived from compiler IRs. PROGRAML is the first representation to capture whole-program control-, data-, and call relations between instructions and operands as well as their order and data types. PROGRAML is a compiler-agnostic design for use at all points in the optimization pipeline; we provide implementations for LLVM and XLA IRs.

- We introduce a benchmark dataset that poses a suite of established compiler analysis tasks as supervised machine learning problems. DEEPDATAFLOW comprises five tasks that require, in combination, the ability to model: control- and data-flow, function boundaries, instruction types, and the type and order of operands over complex programs. DEEPDATAFLOW is constructed from 461k real-world program IRs covering a diverse range of domains and source languages, totaling 8.5 billion data flow analysis classification labels.

- We adapt Gated-Graph Neural Networks (GGNN) to the PROGRAML representation. We show that, within a bounded problem size, our approach achieves $\geq 0.939$ $F_1$ score on all analysis tasks, a significant improvement over state-of-the-art representations. In evaluating the limits of this approach we propose directions to better learn over programs.

## 2 RELATED WORK

Data flow analysis is a long established area of work firmly embedded in modern compilers. Despite its central role, there has been limited work in learning such analysis. Bielik et al. (2017) use ASTs and code synthesis to learn rule-sets for static analyses, some of which are dataflow-related. Our approach does not require a program generator or a hand-crafted DSL for rules. Shi et al. (2020) and Wang & Su (2020) use dynamic information (e.g., register snapshots and traces) from instrumented binaries to embed an assembler graph representation. We propose a static approach that does not need runtime features. Si et al. (2018) use a graph embedding of an SSA form to generate invariants. The lack of phi nodes and function call/return edges means that the representation is not suitable for interprocedural analysis as it stands. Kanade et al. (2020) explore a large-scale, context-dependent vector embedding. This is done at a token level, however, and is unsuited for dataflow analysis.

Prior work on learning over programs employed methods from Natural Language Processing that represented programs as a sequence of lexical tokens (Allamanis, 2016; Cummins et al., 2017a).

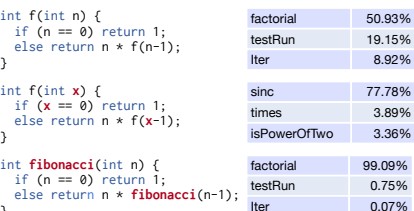 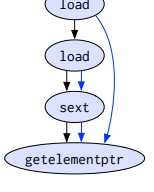 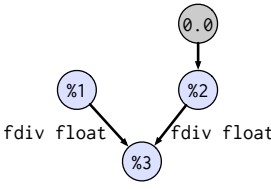

(a) code2vec is sensitive to naming over semantics.

(b) CDFG omits operands.

(c) XFG cannot distinguish non-commutative statements.

Figure 2: Limitations in state-of-the-art learnable code representations: code2vec (Alon et al., 2018a), CDFG (Brauckmann et al., 2020), and XFG (Ben-Nun et al., 2018).

However, source-level representations are not suited for analyzing partially optimized compiler IRs as the input source cannot be recovered. In program analysis it is critical to capture the structured nature of programs (Raychev et al., 2015; Allamanis et al., 2017; Alon et al., 2018b). Thus, syntactic (tree-based) as well as semantic (graph-based) representations have been proposed (Allamanis et al., 2018; Brauckmann et al., 2020). Dam et al. (2018) annotate nodes in Abstract Syntax Trees (ASTs) with type information and employ Tree-Based LSTMs (Tai et al., 2015) for program defect prediction. Both Raychev et al. (2015) and Alon et al. (2018a;b) use path-based abstractions of the AST as program representations, while Allamanis et al. (2017) augment ASTs with a hand-crafted set of additional typed edges and use GGNNs (Li et al., 2015) to learn downstream tasks related to variable naming. Another line of research considers modelling binary similarity via control-flow graphs (CFGs) with an adaptation of GNNs called Graph Matching Networks (Li et al., 2019).

The history of IR-based graph representations for optimization goes back to Ferrante et al. (1987), who remove superfluous control-flow edges to ease optimization with a compact graph representation. A more contemporary precursor to our approach is the ConteXtual Flow Graph (XFG) (Ben-Nun et al., 2018), which combines control-flow with data-flow relations in order to learn unsupervised embeddings of LLVM-IR statements. XFGs omit information that is critical to analysis including the notion of argument order, vertices for both variables and constants, and all control-flow edges. PROGRAML, in combining call-graphs (CG), control-flow graphs, and data-flow graphs (DFG), offers an IR-level program representation that is designed to be useful for a variety of purposes from specific program analyses to downstream optimization tasks. Control and Data Flow Graphs (CDFG) (Brauckmann et al., 2020) use graph vertices for statements and have bi-directional edges for control and data dependencies. The CDFG uses only the instruction opcode to represent a statement, omitting operands, variables, data types, and constants. This prohibits the reasoning about variables and expressions that are required for many data flow analyses, including 3 out of the 5 benchmark tasks that we establish below. Mendis et al. (2019) represent LLVM-IR using a graph that is specialized to a vectorization task. They use unique edge types to differentiate the first five operand positions and augment the graph structure with vectorization opportunities that they compute a priori. Our approach is not specialized to a task, enabling such opportunities learned (e.g., subexpression detection), and uses a embedding weighting to differentiate edge positions without having to learn separate edge transfer weights for each. Finally, an alternate approach is taken by IR2Vec (Keerthy S et al., 2019), an LLVM-IR-specific representation that elegantly models part-of-statements as relations. However, in order to compute the values of the embeddings, IR2Vec requires access to the type of data flow analyses that our approach learns from data alone.

## 3 A GRAPHICAL REPRESENTATION FOR DEEP PROGRAM ANALYSIS

This section presents PROGRAML, a novel IR-based program representation that closely matches the data structures used traditionally in inter-procedural data flow analysis and can be processed natively by deep learning models. We represent programs as directed multigraphs where instructions, variables, and constants are vertices, and relations between vertices are edges. Edges are typed to differentiate control-, data-, and call-flow. Additionally, we augment edges with a local position attribute to encode the order of operands to instructions, and to differentiate between divergent branches in control-flow.

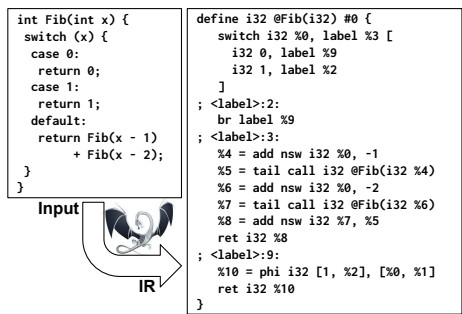

(a) The input program is passed through the compiler front-end to produce an IR. In this example, LLVM-IR is used.

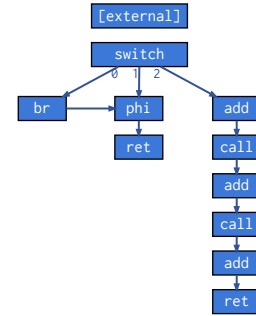

(b) A full-flow graph is constructed of instructions and control dependencies. All edges have position attributes; for clarity, we have omitted position labels where not required.

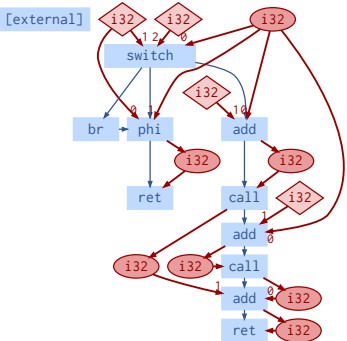

(c) Vertices are added for data elements (elliptical nodes are variables, diamonds are constants). Data edges capture use/def relations. `i32` indicates 32 bit signed integers. Numbers on edges indicate operand positions.

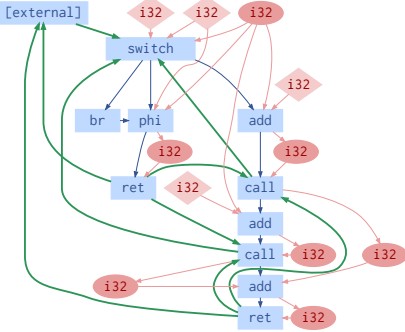

(d) Functions have a single entry instruction and zero or more exit instructions. Call edges are inserted from call sites to function entry instructions, and return-edges from function exits to call sites.

Figure 3: PROGRAML construction from a Fibonacci implementation using LLVM-IR.

We construct a PROGRAML graph $G = (V, E)$ by traversing a compiler IR. An initially empty graph $G = \emptyset$ is populated in three stages: control-flow, data-flow, and call-flow, shown in Figure 3. In practice the three stages of graph construction can be combined in a single $\mathcal{O}(|V| + |E|)$ pass.

**(I) Control Flow** We construct the full-flow graph of an IR by inserting a vertex for each instruction and connecting control-flow edges (Fig. 3a, 3b). Control edges are augmented with a numeric position using an ascending sequence based on their order in the list of an instruction's successors.

**(II) Data Flow** We introduce constant values and variables as graph vertices (Fig. 3c). Data-flow edges are inserted to capture the relation from constants and variables to the instructions that use them as operands, and from instructions to produced variables. As each unique variable and constant is a vertex, variables can be distinguished by their scope, and unlike the source-level representations of prior works, variables in different scopes map to distinct vertices and can thus be discerned. Data edges have a position attribute that encodes the order of operands for instructions. The latent representation of a statement (e.g., `%1 = add i32 %0, 1`) is thus a function of the vertex representing the instruction and the vertices of any operand variables or constants, modulated by their order in the list of operands.

**(III) Call Flow** Call edges capture the relation between an instruction that calls a function and the entry instruction of the called function (Fig. 3d). Return call edges are added from each of the terminal instructions of a function to the calling statement. Control edges do not span functions, such that an IR with functions $F$ produces $|F|$ disconnected subgraphs (the same is not true for data edges which may cross function boundaries, e.g., in the case of a global constant which is used across many parts of a program). For IRs that support external linkage, an additional vertex is

created representing an external call site and connected to all externally visible functions. If a call site references a function not defined in the current IR, a *dummy* function is created consisting of a single instruction vertex and connected through call edges to all call sites in the current IR. A unique dummy function is created for each externally defined function.

## 4 GRAPH-BASED MACHINE LEARNING FOR PROGRAM ANALYSIS

We formulate our system in a Message Passing Neural Network (MPNN) framework (Gilmer et al., 2017). Our design mimics the *transfer functions* and *meet operators* of classical iterative data flow analysis (Kam & Ullman, 1977; Cooper et al., 2004), replacing the rule-based implementations with learnable analogues (message and update functions). This single unified model can be specialized through training to solve a diverse set of problems without human intervention or algorithm design.

The PROGRAML model is an adaptation of GGNN (Li et al., 2015) that takes as input an attributed directed multigraph as presented in Section 3. It consists of three logical phases: input encoding, message propagation and update, and result readout.

**(I) Input Encoding**  Starting from the augmented graph representation $G = (V, E)$, we capture the semantics of the program graph vertices by mapping every instruction, constant, and variable vertex $v \in V$ to a vector representation $h_v^0 \in \mathbb{R}^d$ by lookup in a fixed-size embedding table. The mapping from vertex to learnable embedding vector $f : v \mapsto h_v^0$ must be defined for each IR.

For LLVM-IR, we construct an embedding key from each vertex using the name of the instruction, e.g., `store`, and the data type for variables and constants, e.g., `i32*` (a pointer to a 32-bit integer). In this manner we derive the set of unique embedding keys using the graph vertices of a training set of LLVM-IRs described in Section 5.1. This defines the embedding table used for training and deployment. An *unknown element* embedding is used during deployment to map embedding keys which were not observed in the training data. Since composite types make the size of the vocabulary unbounded in principle, our data-driven approach trades a certain amount of semantic resolution against good coverage of the vocabulary by the available datasets (cf. Table **??**). The embedding vectors are trained jointly with the rest of the model.

**(II) Message Propagation**  Each iteration step is divided into a message propagation followed by vertex state update. Receiving messages $M(h_w^{t-1}, e_{wv})$ are a function of neighboring states and the respective edge. Messages are mean-aggregated over the neighborhood after transformation with a custom position-augmented transfer function that scales $h_w$ elementwise with a position-gating vector $p(e_{wv})$:

$$M(h_w^{t-1}, e_{wv}) = W_{\text{type}(e_{wv})}\left(h_w^{t-1} \odot p(e_{wv})\right) + b_{\text{type}(e_{wv})} \tag{1}$$

The position-gating $p(e_{wv}) = 2\sigma(W_p \, \text{emb}(e_{wv}) + b_p)$ is implemented as a sigmoid-activated linear layer mapping from a constant sinusoidal position embedding  (Vaswani et al., 2017; Gehring et al., 2017). It enables the network to distinguish non-commutative operations such as division, and the branch type in diverging control-flow. In order to allow for reverse-propagation of information, which is necessary for backward compiler analyses, we add backward edges for each edge in the graph as separate edge-types. In all our experiments, we employ Gated Recurrent Units (GRU) (Cho et al., 2014) as our update function.

Step (II) is iterated $T$ times to extract vertex representations that are contextualized with respect to the given graph structure.

**(III) Result Readout**  Data flow analyses compute value sets composed of instructions or variables. We support per-instruction and per-variable classification tasks using a *readout head* on top of the iterated feature extraction, mapping, for each vertex, the extracted vertex features $h_v^T$ to probabilities $R_v(h_v^T, h_v^0)$:

$$R_v(h_v^T, h_v^0) = \sigma\left(f(h_v^T, h_v^0)\right) \cdot g(h_v^T) \tag{2}$$

where $f(\cdot)$ and $g(\cdot)$ are linear layers and $\sigma(\cdot)$ is the sigmoid activation function.

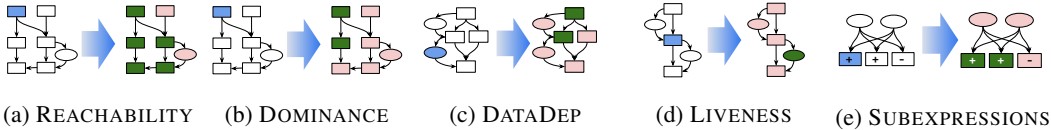

(a) REACHABILITY   (b) DOMINANCE   (c) DATADEP   (d) LIVENESS   (e) SUBEXPRESSIONS

Figure 4: Example input-output graphs for each of the five DEEPDATAFLOW tasks. A single vertex is randomly selected from the input graph as the starting point for computing analysis results, indicated using the *vertex selector* (blue node). Each vertex in the output graph is annotated with a binary value after the analysis has completed. As a supervised classification task, the goal is to predict the output vertex labels given an input graph. These small graphs are for illustrative purposes, the average DEEPDATAFLOW graph contains 581 vertices and 1,051 edges.

## 5   DATA FLOW EXPERIMENTS

Data flow analysis is at the heart of modern compiler technology. We pose a suite of data flow analyses as supervised learning tasks to benchmark the representational power of machine learning approaches. We selected a diverse set of tasks that capture a mixture of both forward and backward analyses, and control-, data-, and procedure-sensitive analyses. *Full details of the analyses are provided in Appendix A.* These particular data flow analyses can already be perfectly solved by non-ML techniques. Here, we use them to benchmark the capabilities of machine learning techniques.

### 5.1   THE DEEPDATAFLOW DATASET

We assembled a 256M-line corpus of LLVM-IR files from a variety of sources and produced labeled datasets using five traditional data flow analyses: control reachability, dominators, data dependencies, liveness, and subexpressions. Each of the 15.4M analysis examples consists of an input graph in which a single vertex is annotated as the root node for analysis, and an output graph in which each vertex is annotated with a binary label corresponding to its value once the data flow analysis has completed (Fig. 4). A 3:1:1 ratio is used to divide the examples for the five problems into training, validation, and test instances. *For further details see Appendix B.*

### 5.2   MODELS

We evaluate the effectiveness of our approach against two contrasting state-of-the-art approaches for learning over programs: one sequential model and one other graph model.

**(I) Sequential Model**   *inst2vec* (Ben-Nun et al., 2018) sequentially processes the IR statements of a program to perform whole-program classification. An IR is tokenized and then mapped into a sequence of pre-trained 200 dimensional embedding vectors which are processed by an LSTM. The final state of the LSTM is fed through a two-layer fully connected neural network to produce a classification of the full sequence. We extend this approach by concatenating to the input sequence a one-hot *token-selector* to indicate the starting point for analysis. Then, we feed the LSTM state through a fully connected layer after every token, producing a prediction for each instruction of the IR. We use the same model parameters as in the original work.

**(II) Graph Models**   We use the model design outlined in Section 4 with two input representations: CDFG (Brauckmann et al., 2020), and PROGRAML. For both approaches we use 32 dimensional embeddings initialized randomly, as in Brauckmann et al. (2020). Input *vertex-selectors*, encoded as binary one-hot vectors, are used to mark the starting point for analyses and are concatenated to the initial embeddings. For CDFG, we use the vocabulary described in Brauckmann et al. (2020). For PROGRAML, we derive the vocabulary from the training set.

Message Passing Neural Networks typically use a small number of propagation steps out of practical consideration for time and space efficiency (Gilmer et al., 2017; Brauckmann et al., 2020). In contrast, data flow analyses iterate until a fixed point is reached. In this work we iterate for a fixed number $T$ of message passing steps and exclude from the training and validation sets graphs for which a traditional implementation of the analysis task requires greater than $T$ iterations to solve.

| | Vocabulary Size | Vocabulary Test Coverage |
|---|---|---|
| inst2vec | 8,565 | 34.0% |
| CDFG | 75 | 47.5% |
| PROGRAML | 2,230 | **98.3%** |

Table 1: Vocabularies for LLVM-IR

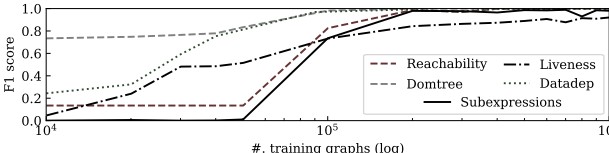

Figure 5: $F_1$ score on a 10k-graph validation set as a function of the number of training graphs.

We set $T = 30$ for training in all experiments and trained a model per task. Once trained, we evaluate model inference using different $T$ values to accommodate programs which required a greater number of steps to compute the ground truth. *See Appendix C.2 for training details.*

## 5.3 EVALUATION

First, we evaluate the effectiveness of each vocabulary at representing unseen programs. Then we evaluate model performance on increasingly large subsets of the DEEPDATAFLOW (DDF) test sets.

**Vocabulary Coverage** Each of the three approaches uses a vocabulary to produce embeddings that describe the instructions and operands of a program. inst2vec uses a vocabulary of 8,565 LLVM-IR statements (where a statement is an instruction and its operands) with identifiers and literals stripped. CDFG uses the 75 LLVM instruction opcodes. For PROGRAML we derive a vocabulary on a set of training graphs that represents instructions and data types separately. Table 1 compares the coverage of these vocabularies as a percentage of the vertices in the test graphs that can be described using that vocabulary. PROGRAML provides $2.1\times$ the coverage on unseen programs as state-of-the-art approaches, the best of which can represent fewer than half of the graph vertices of unseen programs.

**DDF-30: Testing on Limited Problem Size** We initially limit our testing to the subset of each task's test set which can be solved using a traditional analysis implementation in $\leq 30$ steps, denoted DDF-30. This matches the $T = 30$ message passing iterations used in computing the graph models' final states to ensure that a learned model, if it has successfully approximated the mechanism of an analysis, has sufficient message passing iterations to solve each test input. Table 2 summarizes the performance of inst2vec, CDFG, and PROGRAML.

The relational representation of our approach shows excellent performance across all of the tasks. CDFG, which also captures control-flow, achieves comparable performance on the REACHABILITY and DOMINANCE tasks. However, the lack of operand vertices, positional edges, and data types renders poor performance on the SUBEXPRESSIONS task. Neither the CDFG or inst2vec representations enable per-variable classification, so are incapable of the DATADEP and LIVENESS tasks. To simplify comparison, we exclude these two tasks from inst2vec and CDFG aggregate scores. In spite of this, PROGRAML correctly labels $4.50\times$ and $1.12\times$ more nodes than the state-of-the-art approaches. The weakest PROGRAML performance is on the LIVENESS task. When model performance is considered as a function of the number of training graphs, shown in Figure 5, we see that the performance of PROGRAML quickly converges towards near-perfect $F_1$ scores on a holdout validation set for all tasks except LIVENESS, where the model is still improving at the end of training. This suggests estimating the *transfer* (message) and *meet* (update) operators of this backwards analysis poses a greater challenge for the network, and may benefit from further training.

**DDF-60: Generalizing to Larger Problems** The DDF-30 set excludes 28.7% of DEEP-DATAFLOW graphs which require more than 30 steps to compute ground truth labels. To test whether these learned models can generalize to solve larger problems, we used the models we trained at $T = 30$ but double the number of inference message passing steps to $T = 60$ and repeated the tests on all graphs which require $\leq 60$ analysis steps (excluding 19.6%). The results of this experiment, denoted DDF-60, are shown in Table 2. We observe that performance is consistent on this larger problem set, demonstrating that PROGRAML models can generalize to problems larger than those they were trained on. The results indicate that an approximate fixed-point algorithm is learned by the model, a critical feature for enabling practical machine learning over programs.

Table 2: Data flow analysis results. For the restricted subset DDF-30 PROGRAML obtains strong results. Results on the full dataset (DDF) highlight the scalability challenges of MPNNs.

| Analysis | Example Optimization | | inst2vec DDF-30 | CDFG DDF-30 | PROGRAML DDF-30 | PROGRAML DDF-60 | PROGRAML DDF |
|---|---|---|---|---|---|---|---|
| Reachability | Dead Code | Precision | 0.105 | **1.000** | 0.998 | 0.997 | 0.996 |
| | Elimination | Recall | 0.007 | 0.996 | **0.998** | 0.998 | 0.917 |
| | | $F_1$ | 0.012 | **0.998** | 0.998 | 0.997 | 0.943 |
| Dominance | Global Code Motion | Precision | 0.053 | 0.999 | **1.000** | 0.983 | 0.066 |
| | | Recall | 0.002 | **1.000** | **1.000** | 1.000 | 0.950 |
| | | $F_1$ | 0.004 | 0.999 | **1.000** | 0.991 | 0.123 |
| DataDep | Instruction Scheduling | Precision | — | — | **0.998** | 0.992 | 0.987 |
| | | Recall | — | — | **0.997** | 0.996 | 0.949 |
| | | $F_1$ | — | — | **0.997** | 0.993 | 0.965 |
| Liveness | Register Allocation | Precision | — | — | **0.962** | 0.931 | 0.476 |
| | | Recall | — | — | **0.916** | 0.955 | 0.925 |
| | | $F_1$ | — | — | **0.937** | 0.939 | 0.625 |
| Subexpressions | Global Common | Precision | 0.000 | 0.139 | **0.997** | 0.954 | 0.938 |
| | Subexpression | Recall | 0.000 | 0.005 | **0.996** | 0.999 | 0.992 |
| | Elimination | $F_1$ | 0.000 | 0.009 | **0.996** | 0.967 | 0.959 |

**DDF: Scalability Challenges** Finally, we test the analysis models that were trained for $T = 30$ message passing iterations on all DEEPDATAFLOW graphs, shown in Table 2 as DDF. We use $T = 200$ inference message passing iterations to test the limits of stability and generalization of current graph neural networks. 9.6% of DDF graphs require more than 200 steps to compute. Therefore, this experiment highlights two of the challenges in the formulation of data flow analysis in an MPNN framework: first, that using a fixed number of message passing iterations across each and every edge leads to unnecessary work for problems that can be solved in fewer iterations or by propagating only along a dynamic subset of the edges at each timestep (the maximum number of steps required by a graph in DDF is 28,727). Secondly, models that compute correct results for a graph when processed for an appropriate number of steps may prove unstable when processed for an excessively large number of steps. In Table 2 we see substantial degradations of model performance in line with these two challenges. DOMINANCE and LIVENESS show reductions in precision as the models over-approximate and have a large number of false positives. REACHABILITY and DATADEP, in contrast, show drops in recall as the fixed $T = 200$ iterations is insufficient to propagate the signal to the edges of large problems. MPNNs do not scale in the way that we should like for large programs. A part of this, we believe, is that using a generic MPNN system is wasteful. Ordinary data flow engines process nodes in a particular order (usually reverse post order) and are naturally able to identify that a fixed point has been reached. We believe that *dynamically-sparse* message passing strategies and an adaptive number of iterations could address these scalability challenges.

## 6 DOWNSTREAM TASKS

In the previous section we focus on data flow analysis as a benchmark for the capabilities of machine learning for compiler analysis. For the analyses considered, non-ML techniques achieve perfect scores. In this section we apply PROGRAML to two downstream data flow tasks for which non-ML techniques fail: predicting heterogeneous compute device mappings and algorithm classification. In both domains PROGRAML outperforms prior graph-based and sequence-based representations, reducing test error by $1.20\times$ and $1.35\times$, respectively. Finally, we ablate every component of our representation and summarize the contribution of each.

Table 3: Predicting heterogeneous compute device mapping.

| | AMD Error [%] | AMD Precision | AMD Recall | NVIDIA Error [%] | NVIDIA Precision | NVIDIA Recall |
|---|---|---|---|---|---|---|
| Static Mapping | 41.2 | 0.35 | 0.59 | 43.1 | 0.32 | .57 |
| DeepTune | 28.1 | 0.72 | 0.72 | 39.0 | 0.69 | 0.61 |
| DeepTune$_{IR}$ | 26.2 | 0.76 | 0.74 | 31.6 | 0.70 | 0.68 |
| inst2vec | 19.7 | 0.81 | 0.80 | 21.5 | 0.79 | 0.79 |
| PROGRAML | **13.4** | **0.89** | **0.87** | **20.0** | **0.81** | **0.80** |

Table 4: Algorithm classification comparison to state-of-the-art, and ablations.

(a) Comparison to state-of-the-art.

|  | Error [%] | Relative [%] |
|---|---|---|
| TBCNN | 6.00 | +77.5 |
| NCC | 5.17 | +53.0 |
| XFG w. inst2vec vocab | 4.56 | +34.9 |
| XFG | 4.29 | +26.9 |
| PROGRAML | **3.38** | - |

(b) PROGRAML ablations.

|  | Error [%] | Relative [%] |
|---|---|---|
| No vocab | 3.70 | +9.5 |
| inst2vec vocab | 3.78 | +11.8 |
| No control edges | 3.88 | +14.8 |
| No data edges | 7.76 | +129.6 |
| No call edges | 3.88 | +14.8 |
| No backward edges | 4.16 | +23.1 |
| No edge positions | 3.43 | +1.5 |

## 6.1 HETEROGENEOUS DEVICE MAPPING

We apply our methodology to the challenging domain of heterogeneous compute device mapping. Given an OpenCL kernel and a choice of two devices to run it on (CPU or GPU), the task is to predict the device which will provide the best performance. This problem has received significant prior attention, with previous approaches using both hand-engineered features (Grewe et al., 2013) and sequential models (Cummins et al., 2017a; Ben-Nun et al., 2018). We use the OPENCL DEVMAP dataset (Cummins et al., 2017a), which provides 680 labeled CPU/GPU instances derived from 256 OpenCL kernels sourced from seven benchmark suites on two combinations of CPU/GPU hardware, AMD and NVIDIA. *cf. Appendix D.1 for details.*

The performance of PROGRAML and baseline models is shown in Table 3. As can be seen, PRO-GRAML outperforms prior works. We set new state-of-the-art $F_1$ scores of 0.88 and 0.80.

## 6.2 ALGORITHM CLASSIFICATION

We apply our approach to the task of classifying algorithms from unlabeled implementations. We use the Mou et al. (2016) dataset. It contains implementations of 104 different algorithms that were submitted to a judge system. All samples were written by students in higher education. There are around 500 samples per algorithm. We compile them with different combinations of optimization flags to generate a dataset of overall 240k samples, as in Ben-Nun et al. (2018). Approximately 10,000 files are held out each as development and test sets. *cf. Appendix D.2 for details.*

Table 4a compares the test error of our method against prior works.

**Ablation Studies** We ablate the PROGRAML representation in Table 4b. Every component of our representation contributes positively to performance. We note that structure alone (*No vocab*) is sufficient to outperform prior work, suggesting that algorithm classification is a problem that lends itself especially well to judging the power of the representation structure, since most algorithms are well-defined independent of implementation details, such as data types. However, the choice of vocabulary is important. Replacing the PROGRAML vocabulary with that of a prior approach (*inst2vec vocab*) degrades performance. The greatest contribution to the performance of PROGRAML on this task is data flow edges. Backward edges, which are required for reasoning about backward data flow analyses, provide the second greatest contribution. These results highlight the importance of data flow analysis for improving program reasoning through machine learning.

## 7 CONCLUSIONS

The evolution of ML for compilers requires more expressive representations. We show that current techniques cannot reason about simple data flows which are at the core of all compilers. We present PROGRAML, a graph-based representation for programs derived from compiler IRs that accurately captures the semantics of a program's statements and the relations between them. We are releasing the DEEPDATAFLOW dataset as a community benchmark for evaluating approaches to learning over programs. PROGRAML and DEEPDATAFLOW open up new directions for research towards more flexible and useful program analysis. PROGRAML outperforms the state-of-the-art, but is limited by scalability issues imposed by MPNNs. As future work, we will investigate how MPNNs could be improved to learn efficient and stable fixed-point algorithms, regardless of input graph size.

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

# A   DATA FLOW DEFINITIONS

This section provides the definitions of the five analysis tasks used in this paper to evaluate the representational power of deep learning over programs. We chose a diverse set of analysis tasks to capture a mixture of both forward and backward analyses, and control-, data-, and procedure-sensitive analyses.

**(I) REACHABILITY: Reachable Instructions**   Control reachability is a fundamental compiler analysis which determines the set of points in a program that can be reached from a particular starting point. Given $\text{succ}(n)$, which returns the control successors of an instruction $n$, the set of reachable instructions starting at root $n$ can be found using forward analysis:

$$\text{Reachable}(n) = \{n\} \bigcup_{p \in \text{succ}(n)} \text{Reachable}(p) \tag{3}$$

**(II) DOMINANCE: Instruction Dominance**   Instruction $n$ dominates statement $m$ if every control-flow path the from the program entry $n_0$ to $m$ passes through $n$. Like reachability, this analysis only requires propagation of control-flow, but unlike reachability, the set of dominator instructions are typically constructed through analysis of a program's reverse control-flow graph (Lengauer & Tarjan, 1979; Blazy et al., 2015):

$$\text{Dom}(n) = \{n\} \cup \left( \bigcap_{p \in \text{pred}(n)} \text{Dom}(p) \right) \tag{4}$$

Where $\text{pred}(n)$ returns the control predecessors of instruction $n$. We formulate the DOMINANCE problem as: Given a root instruction vertex $n$, label all vertices $m$ where $n \in \text{Dom}(m)$.

**(III) DATADEP: Data Dependencies**   The data dependencies of a variable $v$ is the set of predecessor instructions that must be evaluated to produce $v$. Computing data dependencies requires traversing the reverse data-flow graph:

$$\text{DataDep}(n) = \text{defs}(n) \cup \left( \bigcup_{p \in \text{defs}(n)} \text{DataDep}(p) \right) \tag{5}$$

Where $\text{defs}(n)$ returns the instructions that produce the operands of $n$.

**(IV) LIVENESS Live-out variables**   A variable $v$ is live-out of statement $n$ if there exists some path from $n$ to a statement that uses $v$, without redefining it. Given $\text{uses}(n)$, which returns the operand variables of $n$, and $\text{defs}(n)$, which returns defined variables, the live-out variables can be computed forwards using:

$$\text{LiveOut}(n) = \bigcup_{s \in \text{succ}(n)} \text{uses}(s) \cup \big(\text{LiveOut}(s) - \text{defs}(s)\big) \tag{6}$$

**(V) Global Common Subexpressions**   The identification of common subexpressions is an important analysis for optimization. For compiler IRs we define a subexpression as an instruction and its operands, ordered by either their position (for non-commutative operations), or lexicographically (for commutative operations). We thus formulate the common subexpression problem as: Given an instruction (which forms part of a subexpression), label any other instructions in the program which compute the same subexpression. This is an inter-procedural analysis, though operands must obey their scope. Common subexpressions are typically identified using available expression analysis:

$$\text{Avail}(n) = \text{uses}(n) \cup \left( \bigcap_{p \in \text{pred}(n)} \text{Avail}(p) \right) - \text{defs}(n) \tag{7}$$

Where $\text{uses}(n)$ return the expressions used by instruction $n$, and $\text{defs}(n)$ returns the expressions defined by $n$.

Table 5: The DEEPDATAFLOW LLVM-IR corpus.

| | Language | Domain | IR files | IR lines |
|---|---|---|---|---|
| BLAS 3.8.0 | Fortran | Scientific Computing | 300 | 345,613 |
| GitHub | C | Various | 38,109 | 74,230,264 |
| | OpenCL | | 5,224 | 9,772,858 |
| | Swift | | 4,386 | 4,586,161 |
| Linux 4.19 | C | Operating Systems | 13,418 | 41,904,310 |
| NPB (Bailey et al., 1991) | C | Benchmarks | 122 | 255,626 |
| Cummins et al. (2017a) | OpenCL | Benchmarks | 256 | 149,779 |
| OpenCV 3.4.0 | C++ | Computer Vision | 432 | 2,275,466 |
| POJ-104 (Mou et al., 2016) | C++ | Standard Algorithms | 397,032 | 104,762,024 |
| Tensorflow (Abadi et al., 2016) | C++ | Machine learning | 1,903 | 18,152,361 |
| Total | | | 461,182 | 256,434,462 |

Table 6: Characterization of DEEPDATAFLOW subsets.

| | DDF-30 | DDF-60 | DDF-200 | DDF |
|---|---|---|---|---|
| Max. data flow step count | 30 | 60 | 200 | 28,727 |
| #. classification labels | 6,038,709,880 | 6,758,353,737 | 7,638,510,145 | 8,623,030,254 |
| #. graphs (3:1:1 train/val/test) | 10,951,533 | 12,354,299 | 13,872,294 | 15,359,619 |
| Ratio of full test set | 71.3% | 80.4% | 90.3% | 100% |

## B    DEEPDATAFLOW DATASET

The DEEPDATAFLOW dataset comprises: 461k LLVM-IR files assembled from a range of sources, PROGRAML representations of each of the IRs, and 15.4M sets of labeled graphs for the five data flow analyses described in the previous section, totaling 8.5B classification labels.

**Programs**    We assembled a 256M-line corpus of real-world LLVM-IRs from a variety of sources, summarized in Table 5. We selected popular open source software projects that cover a diverse range of domains and disciplines, augmented by uncategorized code mined from popular GitHub projects using the methodology described by Cummins et al. (2017b). Our corpus comprises five source languages (C, C++, Fortran, OpenCL, and Swift) covering a range of domains from functional to imperative, high-level to accelerators. The software covers a broad range of disciplines from compilers and operating systems to traditional benchmarks, machine learning systems, and unclassified code downloaded from popular open source repositories.

**PROGRAML Graphs**    We implemented PROGRAML construction as an `llvm::ModulePass` using LLVM version 10.0.0 and generated a graph representation of each of the LLVM-IRs. PROGRAML construction takes an average of 10.72ms per file. Our corpus of unlabeled graphs totals 268M vertices and 485M edges, with an average of 581 vertices and 1,051 edges per graph. The maximum edge position is 355 (a large `switch` statement found in a TensorFlow compute kernel).

**Data Flow Labels**    We produced labeled graph instances from the unlabeled corpus by computing ground truth labels for each of the analysis tasks described in Section A using a traditional analysis implementation. For each of the five tasks, and for every unlabeled graph in the corpus, we produce $n$ labeled graphs by selecting unique source vertices $v_0 \in V$, where $n$ is proportional to the size of the graph:

$$n = \min\left(\left\lceil \frac{|V|}{10} \right\rceil, 10\right) \tag{8}$$

Each example in the dataset consists of an input graph in which the source vertex is indicated using the *vertex selector*, and an output graph with the ground truth labels used for training or for evaluating the accuracy of model predictions. For every example we produce, we also record the number of steps that the iterative analysis required to compute the labels. We use this value to produce subsets of the dataset to test problems of different sizes, shown in Table 6.

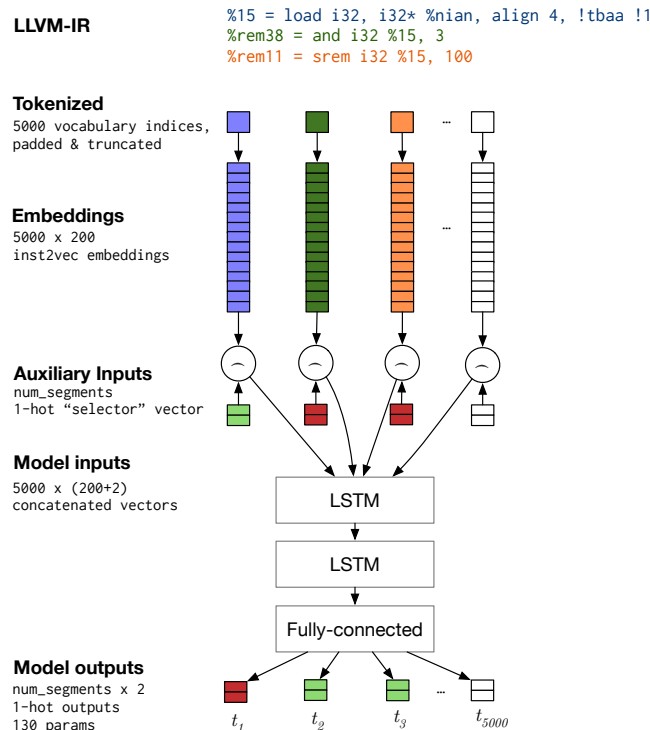

Figure 6: Extending inst2vec (Ben-Nun et al., 2018) to perform per-instruction classification of LLVM-IR. The ⌢ operator denotes vector concatenation.

We divided the datasets randomly using a 3:1:1 ratio for training, validation, and test instances. The same random allocation of instances was used for each of the five tasks. Where multiple examples were derived from a single IR, examples derived from the same IR were allocated to the same split.

As binary classification tasks, data flow analyses display strong class imbalances as only a small fraction of a program graph is typically relevant to computing the result set of an analysis. On the DDF test sets, an accuracy of 86.92% can be achieved by always predicting the negative class. For this reason we report only binary precision, recall, and F1 scores with respect to the positive class when reporting model performance on DEEPDATAFLOW tasks.

## C DATA FLOW EXPERIMENTS: SUPPLEMENTARY DETAILS

This section provides additional details for the experiments presented in Section 5.

### C.1 MODELS

**(I) Sequential Model** The inst2vec model consists of 619,650 trainable parameters in the configuration outlined in Figure 6. The model, implemented in TensorFlow, uses the same parameters as in the original work: $2\times 64$ dimensional LSTM layers followed by a 64 dimensional dense layer and the final 2 dimensional output layer. Sequences are padded and truncated to 5k tokens and processed in batches of 64.

**(II) Graph Models** For CDFG and PROGRAML approaches we use the same model architecture. The model, implemented in PyTorch, consists of a customized GGNN with 87,070 trainable parameters. Batches are implemented as disconnected graphs that are constructed to enable efficient processing of graphs of differing size in parallel without padding. We use a combined batch size of $10,000$ vertices. If a single graph contains more than $10,000$ vertices, it is processed on its own.

Table 7: Average training and inference times on DDF-30 tasks

|  | Train time | Test time | Train time/graph | Val time/graph | Test time/graph |
|---|---|---|---|---|---|
| inst2vec | 10h52m | 1h33m | 45ms | 3ms | 36ms |
| CDFG | 13h14m | 3h27m | 64ms | 1ms | 62ms |
| PROGRAML | 7h21m | 1h39m | 26ms | 3ms | 24ms |

## C.2 EXPERIMENTAL SETUP

**Training Details and Parameters** All models were trained in an end-to-end fashion with the Adam optimizer (Kingma & Ba, 2015) using the default configuration and a learning rate of $1 \cdot 10^{-3}$ for the LSTMs and $2.5 \cdot 10^{-4}$ for the GGNNs. We trained the models on 1M training graphs, evaluating on a fixed 10k validation set at 10k intervals for the first 50k training graphs, and at 100k intervals thereafter. The checkpoint with the greatest validation $F_1$ score is used for testing.

**Runtimes** All experiments were conducted on shared machines equipped with an NVIDIA GTX 1080 GPU, 32GB of RAM, mechanical hard drives, and server-grade Intel Xeon processors. Figure 7 provides measurements of the average runtimes of each approach across the five DDF-30 tasks. In our implementation, we find training and testing to be I/O bound as programs are processed faster than loading many small files from disk. In particular, CDFG performance suffers relative to PROGRAML as the conversion from PROGRAML to CDFG representations is performed on-demand. For validation, inputs are loaded once into system memory and re-used, so the measured time provides a more accurate estimate of processing requirements.

## D DOWNSTREAM TASKS: SUPPLEMENTARY DETAILS

This section provides additional details for the experiments present in Section 6.

### D.1 HETEROGENEOUS COMPUTE DEVICE MAPPING

**Datasets** The OPENCL DEVMAP dataset comprises 256 OpenCL kernels from two combinations of CPU/GPU pairs. The *AMD* set uses an Intel Core i7-3820 CPU and AMD Tahiti 7970 GPU; the *NVIDIA* set uses an Intel Core i7-3820 CPU and an NVIDIA GTX 970 GPU. Each dataset consists of 680 labeled examples derived from the 256 unique kernels by varying dynamic inputs.

**Models** We compare PROGRAML with four approaches: First, with a static baseline that selects the most-frequently optimal device for each dataset (CPU for *AMD*, GPU for *NVIDIA*). Second, with DeepTune (Cummins et al., 2017a), which is a sequential LSTM model at the OpenCL source level. Third, to isolate the impact of transitioning from OpenCL source to LLVM-IR, we evaluate against a new DeepTune$_{IR}$ model, which adapts DeepTune to using tokenized sequences of LLVM-IR as input instead of OpenCL tokens. Finally, we compare against the state-of-the-art approach inst2vec (Ben-Nun et al., 2018), which replaces the OpenCL tokenizer with a sequence of 200-dimensional embeddings, pre-trained on a large corpus of LLVM-IR using a skip-gram model. PROGRAML itself uses the GGNN adaptation as described in the paper. We adapted the readout head to produce a single classification label for each graph, rather than per-vertex classifications, by aggregating over the final iterated vertex states. We also included the available auxiliary input features of the DEVMAP dataset. The auxiliary features are concatenated to the features extracted by the GGNN before classification following the methodology of Cummins et al. (2017a).

The experimental results in this section come from an earlier development iteration of PROGRAML which deviates from the method described in the main paper in the way in which it produces initial vertex embeddings. Instead of deriving a textual representation of instructions and data types to produce a vocabulary, the vocabulary used for the DEVMAP experiment is that of inst2vec (Ben-Nun et al., 2018), where variables and constants are all represented by a single additional embedding vector. The poor vocabulary coverage achieved by using inst2vec motivated us to provide the improved vocabulary implementation that we describe in the main paper (see Table 1).

**Training Details and Parameters**    All neural networks are regularized with dropout (Hinton et al., 2012) for generalization and Batch Normalization (Ioffe & Szegedy, 2015) in order to be uniformly applicable to vastly different scales of auxiliary input features. We used 10-fold cross-validation with rotating 80/10/10 splits by training on 80% of the data and selecting the model with the highest validation accuracy, setting aside $1/10$th of the training data to use for validation. We trained each model for 300 epochs and selected the epoch with the greatest validation accuracy for testing. Baseline models were trained with hyperparameters from the original works. For the PROGRAML results we used 6 layers in the GGNN corresponding to 6 timesteps of message propagation, while sharing parameters between even and odd layers to introduce additional regularization of the weights. We ran a sweep of basic hyperparameters which led us to use the pre-trained inst2vec statement embeddings (Ben-Nun et al., 2018) and to exclude the use of position representations. Both of these hyperparameter choices help generalization by reducing the complexity of the model. This is not surprising in light of the fact that the dataset only contains 680 samples derived from 256 unique programs. PROGRAML was trained with the Adam optimizer with default parameters, a learning rate of $10^{-3}$ and a batch size of 18,000 nodes (resulting in ca. 12000 iteration steps of the optimizer). For the PROGRAML result, we repeat the automated sweep for all hyperparameter configurations and picked the configuration with the best average validation performance. Performance on the unseen tenth of the data is reported.

## D.2    ALGORITHM CLASSIFICATION

**Dataset**    We use the POJ-104 dataset (Mou et al., 2016). It contains implementations of 104 different algorithms that were submitted to a judge system. All programs were written by students in higher education. The dataset has around 500 samples per algorithm. We compile them with different combinations of optimization flags to generate a dataset of overall 240k samples, as in Ben-Nun et al. (2018). Approximately 10,000 files are held out each as a development and test set.

**Models**    We compare with tree-based convolutional neural networks (TBCNN) (Mou et al., 2016) and inst2vec (Ben-Nun et al., 2018). We used author-provided parameters for the baseline models. For PROGRAML we used 4 layers in the GGNN corresponding to 8 timesteps. To further test the expressive power of the graph-based representation against the tree-based (TBCNN) and sequential (inst2vec) prior work, we additionally compare against graph-based baselines based on XFG (Ben-Nun et al., 2018).

To better understand the qualitative aspects of replacing a graph-based representation that captures program semantics like Contextual Flow Graphs (XFG) (Ben-Nun et al., 2018) with the more complete PROGRAML representation, we adapted a GGNN (Li et al., 2015) to directly predict algorithm classes from XFG representations of the programs. In contrast to this, Ben-Nun et al. (2018) used XFG only to generate statement contexts for use in skip-gram pre-training. Here, we lift this graphical representation and make it accessible to a deep neural network directly, as opposed to the structure-less sequential approach in the original work (inst2vec).

**Training Details and Parameters**    All models were trained with the AdamW (Loshchilov & Hutter, 2019) optimizer with learning rate $2.5 \cdot 10^{-4}, \beta_1 = 0.9, \beta_2 = 0.999, \varepsilon = 10^{-8}$ for 80 epochs. Dropout regularization is employed on the graph states with a rate of 0.2.

