# OpenReview forum: "Deep Data Flow Analysis"
_ICLR.cc/2021/Conference — Reject_

### Official Review · AnonReviewer2 · 2020-10-15
**Valuable contribution, some lingering reservations**

**Rating:** 7
**Confidence:** 3

**Review:**

The authors present a language-independent, compiler-agnostic graph representation of programs (ProGraML) designed for machine learning.  Their representation is noteworthy for including the program's control, data, and call dependencies, and more generally for including enough information for standard dataflow analyses to be possible in principle.  Their goal is to train machine learning systems on this representation to provide heuristics to replace manual heuristic-engineering inside compilers.  The authors introduce a large benchmark dataset that poses several established compiler analysis tasks as supervised learning problems, and show that thanks to their well-designed program representation, established network architectures (i.e. Gated Graph Neural Networks) perform well on all analysis tasks.  Perhaps more importantly though only a side note in the text, they show that GGNNs on ProGraML outperform prior approaches on two downstream tasks for which non-ML techniques are not perfect: device mappings and algorithm classification.

I think the paper addresses an important problem, makes a valuable contribution, and could be of broad interest to the community.  Thus I recommend that it be accepted. However, I have lingering reservations about their approach that I hope the authors will consider.  In the 'Heterogeneous Compute Device Mapping' experiment, the message passing is only iterated 6 times, compared to 30 for all dataflow analysis tasks, suggesting that---contra what seems to be implied by the paper---the network's ability to perform such dataflow analyses when T is large might not be a relevant factor in its strong performance on the downstream task.  I am unable to find in the paper the number of rounds used for the algorithm classification experiment.

I think the paper could be strengthened by putting more emphasis on downstream tasks and by addressing the null hypothesis that the network is not doing anything like dataflow analysis for them.  This hypothesis might be addressed by plotting performance as a function of the number of message passing rounds; if performance continues to increase as T is increased, it would constitute some evidence it is using some analogue of dataflow analysis, while if performance peaks or plateaus at a small T, it would constitute some evidence that it is not.  Whether or not it is performing its own dataflow analysis, a second null hypothesis might be that the network could be even better if it simply saw the precomputed dataflow analyses as additional inputs. That is, standard analyses may suffice, and it may not be necessary to give the network the freedom to invent its own analyses.  Needless to say, just because GNNs can recapitulate standard dataflow analyses does not necessarily mean that they should need to recapitulate these analyses, particularly since (a) they are not 100% accurate and (b) they are wasteful and difficult to scale (as discussed in 'DDF: Scalability Challenges').

---

> ### Author Response · Authors · 2020-11-21
> **Reviewer #5 - Author Response**
>
> Many thanks for the positive review and insightful questions! We like that you find our work addresses an important problem, makes a valuable contribution, and could be of broad interest to the community.
>
> 1. Downstream tasks are only a side note.
>
> We agree, we failed to emphasise the importance of the practical applications of our approach to downstream tasks. We have uploaded a revised submission which restructures the text to better communicate the message.
>
> 2. You used T=6 for the DevMap task, but T=30 for data flow tasks. Why?
>
> Great question. We performed a basic parameter sweep and found that T=6 using a six-layer GGNN provided a 13.9% improvement in validation performance over T=30 using a single-layer GGNN. We discuss in the paper that T is a sensitive parameter for the data flow tasks, and we find this to be consistent for downstream tasks. However, the two types of task are fundamentally different. For data flow tasks, the function we want to learn is simple; the challenge is to ensure that the representation is sufficiently expressive and to propagate the results to all affected program points. Hence, a shallow, single-layer network is sufficient, and T=30 is an arbitrary value we selected to ensure that a broad range of problems can be addressed while keeping within the scaling constraints that we discuss in the paper.
>
> For DevMap, the function we want to learn is complex, and is computed over the entire program. Here was see that additional layers aid in the model's capacity to estimate the function, and the node states after iterating for T=6 are aggregated to produce a program embedding.
>
> 3. Missing T value for algorithm classification task.
>
> Sorry about that, we have added this to the revised submission. We used T=8 steps distributed over four layers. As with DevMap, we found that a deeper network and fewer timesteps provided stronger performance on a validation set than a shallow network with many steps.
>
> 4. Null hypothesis: the network is not doing anything like dataflow analysis on downstream tasks.
>
> That is a very interesting suggestion that strikes at the heart of the challenge in writing optimizing compilers! For many optimization tasks, we do not _know_ the analyses that are required to produce the best results, and we must resort to heuristics. It is not sufficient to be good at dataflow. If we only wanted to solve dataflow analyses, we could come up with a much more simple solution. However, we do know that almost every optimization decision within a compiler starts with dataflow analysis. So it stands to reason that for machine learning techniques to succeed at compiler optimization, they too should be able to reason about dataflow. In this work we put existing ML approaches to the test, demonstrate their limitations, propose methods to overcome them, and finally include downstream tasks as supporting evidence for why we should care about dataflow analysis. We think that our work is a big step in the right direction, but there is still work to be done.
>
> 5. Null hypothesis: the network would perform better if fed with precomputed dataflow analyses.
>
> This is the approach taken in IR2Vec (Venkatakeerthy et. al. 2019). In that work, the authors augment a program representation with the results of use-def and reaching definitions analyses, showing improved downstream performance on two optimization tasks. The issue is that there is an unbounded number of potential data flow analyses and, in general, we cannot know what will be important to include as input features for a particular task. In the case of the analysis tasks used in this paper, yes, we could augment the node representations with analysis results, though this is non-trivial. The size of the value sets of these analyses is proportional to the size of the program. We will add a discussion to the paper.

---

### Official Review · AnonReviewer1 · 2020-10-27
**Official Blind Review #1**

**Rating:** 6
**Confidence:** 4

**Review:**

Summary
-------------
A methodology for learning representations of programs using graph neural networks applied to graphs extracted from a compiler intermediate representation is presented. The graph representation captures both control flow as well as data dependencies and also represents calls to/returns from functions. A large new dataset to evaluate if models can learn to perform dataflow analyses is introduced, and experimental results show that the described method can be trained to perform these with high accuracy (at least for small programs). Further experiments in the appendix illustrate that the proposed graph structure is also useful for more standard tasks that cannot be solved well using existing algorithms.

Strong/Weak Points
-------------
* (+) The graph representation is novel and well-explained.
* (+) The experimental results on data-flow tasks show that the representation and model are clearly able to capture important facts about programs.
* (-) I'm surprised by the lack of "real" tasks in the main text of the paper, as it means that the main experiments are not on well-studied tasks. To be honest, I'd prefer just swapping the DDF-60/DDF results + discussion with one of the tasks from appendix D.
* (-) I found the experimental details a bit hard to follow, for example, it remains unclear to me if the models are trained on all tasks together at the same time, or if they are trained separately for each task.

Recommendation
-------------
I think this is nice work, but that the paper could still be significantly improved by reporting more details and potential new experiments. It passes the bar for acceptance as-is, but could be substantially more useful to other researchers if more experiments are described.

Questions
-------------
* While the appendix gives statistics about how many programs in the corpus come from different source languages, it remains unclear if the source language has an influence on the success of the method. Did you analyse the model results by language?
* See above - it wasn't clear to me if one model is trained for each data flow task, or if one model is trained for all of them (with different heads). If the former, why not the latter? If the latter, how does it compare with results of the former?
* Following on this line of thought: Did you evaluate if using the dataflow tasks as a pretraining routine for your model makes sense? I'm envisioning a scenario if you take the fully trained model for the data flow tasks and re-head it for the algorithm classification / placement tasks.

Detail Feedback
-------------
* page 3: "prohibt" -> "prohibit"

---

> ### Author Response · Authors · 2020-11-21
> **Reviewer #4 - Author Response**
>
> Thank you for the positive review. We appreciate that you found the representation novel, and that it captures important facts about programs.
>
> 1. Include downstream tasks in the main paper.
>
> Great suggestion, thanks! We have made this change in the revised submission. Actually, the extra page allowance meant that we were able to move the downstream tasks into the main paper whilst keeping the extended discussion of scalability challenges in the DDF results, which we feel will be of interest to the ICLR audience.
>
> 2. Were models trained on all tasks at the same?
>
> Sorry, this was not clear. We trained one model for each task. Our goal is to show that our representation is sufficiently expressive for a range of analysis tasks, and training a model for each task in turn allows us to control for the different iteration counts required for traditional analyses over each program. It will be non-trivial to merge these datasets, but we really like this suggestion and will explore training a model on all analyses simultaneously for the camera-ready version.
>
> 3. Does the source language affect model accuracy?
>
> No, we found no correlation between the original source language and the performance of data flow tasks. We do note interesting differences between the source languages in our dataset, but none that affect task performance. E.g. OpenCL sources tend to contain fewer instructions and less diverging control flow than C++. We include a granular characterization of programs in our open source release.
>
> 4. Did you use dataflow as pre-training for downstream tasks?
>
> The data flow tasks are highly specialized to test different components of our representation. As is, we do not foresee much opportunity for applying pre-training to the more general downstream tasks. We did investigate unsupervised pre-training using a masked language model objective. This provided a modest (1.01%) performance improvement on the DevMap task, which we omitted for space.

---

### Official Review · AnonReviewer3 · 2020-10-29
**Novelty and usability issues**

**Rating:** 4
**Confidence:** 4

**Review:**

## Summary:

The paper proposes a new program representation to find semantic code embeddings (ProGraML). The authors use these embeddings to perform a number of traditional dataflow analysis tasks that you can find in modern day compilers. When trained under a supervised learning setting, Deep Data Flow (DDF) analysis learns to perform these tasks more accurately compared to other deep program representations.

## Strengths:

* Position-gate mapping to distinguish between positional arguments, branch directions etc.
* Impressive scale of the dataset collected that require considerable engineering work.
* Evaluation on a number of classic dataflow analysis tasks from compiler literature.

## Weaknesses and Questions for authors:

### Novelty and missing related work
* The authors claim that ProGaML is the first to use operand order and data types to come up with an embedding for whole-programs targeting a compiler IR. Important related work is missing. For instance in [1], Mendis et. al. learn a program embedding for a program optimization task that explicitly models operand order and data types for whole functions at the LLVM IR level. [2] uses both dynamic and symbolic information to arrive at a program embedding that captures program semantics. Wang and Su use this embedding to perform the algorithm classification task.

### Scalability and Usability
* How fast is DDF compared to traditional dataflow analysis? If DDF is far slower, then its usability may be limited inside a traditional compiler, especially if traditional dataflow analysis gives more precise results.
* What is the largest graph in the dataset and how long does a typical DDF task take on that graph for T=30? What is the comparative time (wall-clock) for traditional dataflow analysis?
* Did you see any improvement of precision over traditional dataflow analysis when using fewer propagation steps? For example, say traditional  dataflow analysis takes T=2000 steps to reach a fixed point. Can DDF outperform traditional dataflow analysis at T=30 or at a similar step?
* Did you try reusing any embedding trained on one task (task1) as a pre-trained embedding for another task (task2)? If yes, what are the results?  Do the embeddings generalize with minimal finetuning?

### Correctness Guarantees
* I did not see any discussion about guarantees of correctness of DDF. Will there be any violations? For example can there be false reaching definitions. In other terms is DDF sound? I assume F1 score is only measuring preciseness of the analysis in your case. If not, please explain, how exactly you compute them.

[1] Mendis et. al. Compiler auto-vectorization with imitation learning. In Advances in Neural Information Processing Systems 32 (NeurIPS 2019)

[2] Ke Wang and Zhendong Su. 2020. Blended, precise semantic program embeddings. In Proceedings of the 41st ACM SIGPLAN Conference on Programming Language Design and Implementation(PLDI 2020)

---

> ### Author Response · Authors · 2020-11-21
> **Reviewer #3 - Author Response**
>
> Thank you for the valuable feedback! We are encouraged that you appreciate our evaluation using classic data flow analyses, the position-gate mapping, and the scale of our dataset.
>
> It is apparent to us that we failed to communicate the main goal of our work: by increasing the expressivity of program representations (evaluated using data flow analyses as a benchmark) we improve performance on practical downstream tasks. We have restructured our submission to more clearly communicates this message.
>
> 1. Missing related work.
>
> Great suggestions, we added a discussion in the main text, and elaborate here:
>
> "Compiler auto-vectorization with imitation learning" (NeurIPS'19).
>
>  * Operand order: This work proposes an elegant solution for differentiating operand positions by mapping data edges to different edge types based on their position. Positions 1-5 map to five unique edge types, and positions beyond 5 are collapsed to a sixth type. The position-mapping approach we propose scales to larger positions (our dataset contains programs with 355 edge positions) and is applied to control flow to differentiate branches. Mendis et. al.'s representation omits control flow, making it unsuitable for some analysis tasks (e.g. dominators, reachability).
>  * Data types: Mendis et. al. specialize their representation to the vectorization task by adding packing edges between compatible data types that are determined a priori. Our representation is not specialized to a particular task. We represent instruction and data types through node embeddings. This enables relations such as data type compatibility to be learned (e.g. subexpressions task).
>
> "Blended, precise semantic program embeddings" (PLDI'20).
>
>  * Wang & Su combine syntactic-level code representations with execution traces to produce a program embedding. This improves performance relative to syntactic-based representations, but requires executing an instrumented version of the program multiple times with different inputs to gather sufficient traces. Our approach does not require execution and abstractly represents all possible execution paths through control flow edges. Their approach is evaluated on a different algorithm classification dataset (10 classes, 85k instances vs the 104 classes, 240k instances we use) and, as best we can tell, the LiGer source code is not publicly available to evaluate against.
>
> 2. How fast is DDF compared to traditional dataflow analysis? Does this limit usability?
>
> Our goal is not to be faster than traditional dataflow analysis, we haven’t optimized for speed. We use data flow analysis to benchmark the capabilities of ML to reason about programs. We feel this provides a principled means to evaluate our representation that we then apply to downstream tasks.
>
> For downstream tasks, the cost of  producing labels is significantly more expensive than even a slow ML technique.
>
> Producing a label for a program in DevMap takes on average 3.205s on AMD and 3.866s on NVIDIA (measured using the kernel execution time from running each benchmark 5 times, excluding overheads). Predicting the labels with ProGraML takes 24ms (including overheads), 147x faster.
>
> For algorithm classification, the cost of labelling an algorithm by hand is unknown, but it is likely that even the slowest ML techniques would be faster.
>
> 3. What is the largest graph in the dataset and how long does a typical DDF task take on that graph for T=30?
>
> The largest graph in the dataset contains 124k nodes. We do not have per-graph timings, but the average across the entire dataset is 0.79 ms / graph for traditional data flow analyses, and 24 ms / graph for ProGraML. ProGraML is 30.4x slower than the traditional analysis.
>
> Again, our goal is not to be faster than existing implementations of dataflow analyses. Our goal is to use data flow analyses to evaluate ML approaches.
>
> 4. Did you see any improvement of precision over traditional dataflow analysis when using fewer propagation steps?
>
> No we didn’t. We suspect that highway edges or supernodes could provide shortcuts to estimating analysis results in fewer timesteps, but we did not design for this. We designed our representation to resemble that of traditional compiler representations; as such it has the same scaling properties.
>
> 5. Did you try using an embedding pre-trained for one dataflow analysis for another?
>
> That is an interesting suggestion. The data flow tasks are highly specialized and were selected to test different components of our representation. We suspect that there will be little opportunity for transfer learning between them. On the other hand, we could imagine training on a more general task (e.g. algorithm classification) and then specializing to a data flow task, though the margin for improvement is slim. We will discuss in camera-ready.
>
> 6. Correctness guarantees?
>
> No, our approach is not sound. We use F1 score to aggregate prediction performance over binary analysis outputs: `F1 = 2*(prec * recall)/(prec + rec)`.

---

### Official Review · AnonReviewer4 · 2020-10-30
**Novel Program Representation**

**Rating:** 7
**Confidence:** 3

**Review:**

Summary

This work proposes a new program representation called ProGRamL, which is suitable to be used as an input for graph-based machine learning techniques, and which can be used to learn compiler heuristics.  By formulating several problems with graph neural networks, authors evaluate ProGRamL’s ability to represent problems for classic compiler dataflow problems.

Strengths:
* Advances state of the art for representations of programs for ML
* New benchmark dataset for program representation
* Well written, clear paper

Weaknesses:
* Could be viewed as modest tweaks to existing representations.
* Models are sensitive to the number of iterations, indicating they are perhaps not learning the same functions as in the exact algorithm

Comments:

Overall I enjoyed reading this paper and I would be in favor of acceptance.  I very much appreciate the insight that GNNs can essentially learn classic dataflow operators, provided that you can run the GNN for more number of iterations (that seems to be true to some extent, barring caveats provided by the authors).  The representation seems substantially more compute while still being suited for machine learning.  The techniques for connecting this with graph neural networks are also valuable.  The benchmark dataset is appreciated and can be useful for future studies.  The case studies on downstream tasks are also a nice addition that proves this exercise is not purely academic.

One thing to mention is that there does seem to be quite a large abstraction gap between the problems that ProGRamL is designed for (computing low-level node properties), and where it can actually be useful (where we need heuristics due to complex interactions like the algorithm classification and performance prediction).  In that sense, one wonders whether being able to compute dataflow properties is a novelty, or whether it has led to some new innovations that are useful in developing heuristics.  Either way, I think this work is still interesting, pushes the state of the art, and will ultimately provide value for future compiler developers.

I also think it would be nice if the authors can provide more information about the limitations of the representation, and describe when it could conceivably introduce errors/approximations.

Questions:
* One problem with the existing approach is that it is not stable to the number of iterations.  Do you foresee this as a sensitive parameter in downstream tasks?


* Does the representation capture commutativity in operations which are commutative?  (also, does the operand order on the switch really matter?)

* Why does CDFG have such poor vocabulary coverage?

* What’s the reasoning behind using a dummy function to represent an external unavailable function?  Why not a new node type?  Could it be confounding to the analysis to assume a call to an empty function?

---

> ### Author Response · Authors · 2020-11-21
> **Reviewer #2 Author Response**
>
> Thank you very much for the positive review and insightful questions. We are encouraged that you find our paper well written and advances the state of the art. Specific responses below:
>
> 1. Could be viewed as modest tweaks to existing representations.
>
> Yes, we build upon well-established techniques from compiler research. We feel the contributions that will be of interest to the ICLR community are the novel extensions to representing whole-program semantics for GGNNs, the reusable benchmark dataset, and the limitations in MPNN scalability we identify.
>
> 2. Models are sensitive to the number of iterations, indicating they are perhaps not learning the same functions as in the exact algorithm.
>
> You're right, we discuss this in the text, and we think there’s more work to be done. That's why we propose the benchmarks. We think that our representation is a big step in the right direction, but further work on modeling could improve the reasoning. We hope to expedite this new line of research through the release of our benchmark datasets.
>
> 3. There is a large abstraction gap between low-level representation and downstream tasks. Is dataflow just a novelty?
>
> For downstream tasks, we know that something is missing because existing ML approaches don’t achieve perfect accuracy. It could be the program representation, benchmark coverage, dynamic features, noise in the dataset, ... we cannot be sure which. What we do know is that nearly every optimization decision within compilers starts with data flow analysis. In this paper we focus on the program representation and show that existing approaches are insufficient for even some basic data flow tasks. We then show that addressing these limitations improves performance on downstream tasks.
>
> 4. More information about the limitations of the representation.
>
> Thanks for the suggestion, we will expand the discussion of limitations. Briefly: limitations of representation: we omit literals (can’t do const prop or flow-sensitive analyses), we do not achieve perfect vocabulary coverage, and have limited reasoning about composite types. Limitations of model: can’t handle arbitrary sized graphs, non-separable dataflows, and our approach doesn't address the challenge of dynamic features.
>
> 5. Do you foresee T as being a sensitive parameter for downstream tasks?
>
> Yes. In a parameter sweep of the DevMap downstream task we saw a 13.9% range in test error by varying T. This is consistent with our findings on the DDF benchmark using large T values. We think that this is an interesting problem for future work, possibly through extending MPNNs to support dynamic, learned iteration counts.
>
> 6. Do you capture commutativity?
>
> Yes. The subexpressions task includes both commutative and non-commutative expressions. ProGraML achieves an F1 of 0.996 on this task . We were impressed by the ability to learn to differentiate commutative/noncommutative expressions from data alone, and, to the best of our knowledge, this is the first graph representation to enable this.
>
> 7. Does the branch order of a switch really matter?
>
> Branch order is important for flow-sensitive analyses, but we do not yet have a flow-sensitive task in our benchmarks. We are looking to add as one an extension to our open source release.
>
> 8. Why does CDFG have poor vocab coverage?
>
> The CDFG representation omits const/var types.
>
> 9. Why not declare a new node type for unknown functions? Could be confounding to the analysis to assume a call to an empty function?
>
> Great suggestion. We could experiment with this. We chose to use a single instruction since we know that an undefined function must contain at least one instruction, we just don’t know what it is. The same is true for the instruction representing a call from an external call site.

---

### Official Review · AnonReviewer5 · 2020-11-10

**Rating:** 7
**Confidence:** 3

**Review:**

# Changes after author response
Thanks for addressing the concerns in the review with the new revision. I am revising the score to 7 from 5 based on the reply and the revisions to the paper.

---
# Summary
This paper describes a directed graph representation for programs and a graph neural network architecture that can operate on these graphs to create per-vertex features and support per-instruction or per-variable classification tasks. The authors collect a dataset of programs from the web and convert them into the graph format for an empirical evaluation. The paper provides the results of an empirical evaluation using this dataset on several classical dataflow analysis tasks, like graph reachability and variable liveness.

# Strengths
- The graph representation builds upon the work done in LLVM to analyze and simplify programs. This approach contrasts with other previous works that use source code-based representations, and allows the learning portion to build upon a more convenient representation without needing to learn how to duplicate the same kinds of transformations performed by LLVM.
- The empirical results show that the proposed representation and learning method can work better than baseline methods.

# Weaknesses
- Although the method is compared to prior work, there is no ablation of the neural network architecture or details of the graph representation (except for one experiment in the appendix), which would allow us to evaluate the contributions of different parts in a more controlled manner.
- The representation used is not particularly novel, since it can be straightforwardly constructed from LLVM IR.
- The tasks considered have existing exact hand-written solutions, so it is not clear how the methods proposed on the paper can have practical impact.

# Recommendation
Overall, I thought that the approach of leveraging LLVM IR makes a lot of sense, as well as preserving and effectively communicating its structure by using a graph neural architecture. However, it was strange to me that the focus of the empirical evaluation is on dataflow analysis tasks like computing reachability, when these tasks have well-known solutions (in fact, written as one equation each in the appendix). It is good to know that the proposed method can perform these tasks, but that has no direct practical implications since there are already good solutions for these tasks. These empirical results do not assure us that the method will perform well on other, more interesting tasks, either.
I see that there are downstream tasks evaluated in the appendix. I would recommend reorganizing the paper to make these results more prominent, and making sure to compare with the latest work in a fair way. It would also be exciting to see how the paper's methods can be used to other tasks relevant for compilers, such as code optimization-related tasks which would be amenable to a reinforcement learning approach. I would be willing to revise the paper's rating given sufficient revision.

# Questions
- Was there any deduplication performed in the data collection process, to ensure that the same files, functions, or other large snippets would not be present in both the train and test sets? The same file might have been copied into multiple GitHub repositories, for example.
- Why are all external functions references assigned to the same dummy function? That seems to prevent distinguishing between different standard library functions effectively, for example.

---

> ### Author Response · Authors · 2020-11-21
> **Reviewer #1 Author Response**
>
> Thank you very much for the valuable feedback! We have uploaded a revised submission that we hope addresses your concerns. Please see specific responses below:
>
> 1. Difficult to evaluate the contributions of each component of the representation.
>
> Good suggestion. We have moved the ablation study from the appendix into the main text (Table 4b, revised submission), and expanded our discussion. We chose to ablate a downstream task rather than a data flow task since the data flow tasks are highly specialized to test specific components of program representations. For example, dominator analysis is insensitive to operand order, so we foresee no insight from ablating the contribution of edge positions on this task. For downstream tasks we do not know which analyses (and thus which components of our representation) are significant. Our ablation shows that all components contribute positive to performance, to differing degrees.
>
> 2. The representation is not particularly novel.
>
> While the data structures that our representation is derived from (control flow graph, data flow graph, call graph) are well established, the union of all three as a whole-program ML representation is novel, as is the extension of MPNNs to preserve local operand order. Our ablation study (Table 4b, revised submission) supports the positive contribution that each novel contribution makes to performance on downstream tasks.
>
> 3. Doesn't improve over hand-written data flow solutions, practical impact unclear.
>
> You're right. Our goal is not to improve over hand-written solutions for existing data flow analyses. We use those analyses to benchmark the capabilities of machine learning approaches. We think that the limitations we observe in MPNN scalability makes our dataset valuable to the research community, but the practical impact is achieved through increased performance on downstream tasks. We have worked to communicate this message more clearly in the text.
>
> 4. Reorganize the paper to emphasize practical downstream applications.
>
> Great suggestion, thanks! We have done this in the revised submission.
>
> 5. Other downstream tasks? Amenable to reinforcement learning?
>
> We agree. We are working on this. Each new task is a lot of engineering work and squeezes our page count further. We hope that the seven tasks we present in this paper are sufficient, but our software is open source and is constantly being extended. We are encouraged that users are adapting our approach to downstream tasks that we have not yet considered, and are excited to see their progress.
>
> 6. Did you dedupe the dataset?
>
> Yes, we deduped at the file level, so that things like copied or forked code on GitHub is removed, as per your example. We didn’t do granular function/snippet-level deduplication.
>
> 7. Why use a single dummy function for all unknown functions?
>
> Sorry, this wasn’t clear. We create one dummy function per undefined function signature. E.g. `printf()` and `mmap()` will be represented by different dummy functions. All calls to `mmap()` will resolve to the same dummy function.

---

### Decision · Program_Chairs · 2021-01-07
**Final Decision**

**Decision:**

Reject

**Comment:**


After reading the paper, reviews and authors’ feedback. The meta-reviewer agrees with the reviewers that the paper presented a very interesting idea and empirical studies.  R3 rightfully pointed out the need to clarify relation to related works, as well as the scalability issue.

Notably, because the analysis does not ensure correctness, it has limited applicability in tasks where absolution correctness are required(e.g. Dead code), but can benefit downstream tasks that do not require absolute correctness. A more thorough discussion about this perspective would strengthen the paper.

Right now the paper is borderline, the meta-reviewer acknowledges the pros of the paper as mentioned in the reviews, but also thinks the paper can be further improved based on the comment. Therefore the meta-reviewer decided to not accept the paper but would encourage the authors to improve the paper per comments for a future submission.

Thank you for submitting the paper to ICLR.